# Multivariate Time Series Anomaly Detection by Capturing Coarse-Grained Intra- and Inter-Variate Dependencies

## Abstract

Multivariate time series anomaly detection is essential for failure management in web application operations, as it directly influences the effectiveness and timeliness of implementing remedial or preventive measures. This task is often framed as a semi-supervised learning problem, where only normal data are available for model training, primarily due to the labor-intensive nature of data labeling and the scarcity of anomalous data. Existing semi-supervised methods often detect anomalies by capturing intra-variate temporal dependencies and/or inter-variate relationships to learn normal patterns, flagging timestamps that deviate from these patterns as anomalies. However, these approaches often fail to capture salient intra-variate temporal and inter-variate dependencies in time series due to their focus on excessively fine granularity, leading to suboptimal performance. In this study, we introduce MtsCID, a novel semi-supervised multivariate time series anomaly detection method. MtsCID employs a dual network architecture: one network operates on the attention maps of multi-scale intra-variate patches for coarse-grained temporal dependency learning, while the other works on variates to capture coarse-grained inter-variate relationships through convolution and interaction with sinusoidal prototypes. This design enhances the ability to capture the patterns from both intra-variate temporal dependencies and inter-variate relationships, resulting in improved performance. Extensive experiments across seven widely used datasets demonstrate that MtsCID achieves performance comparable or superior to state-of-the-art benchmark methods.

## Keywords

Time Series, Anomaly Detection, Deep Learning, AIOps

## 1 Introduction

Modern society increasingly relies on web-based application systems integrated into distributed systems, cloud computing platforms, and Internet of Things (IoT) devices [18, 22, 33]. These systems function across various sectors, including finance, transportation, telecommunications, media, and industrial operations. Downtime or service interruptions in these critical infrastructures can disrupt daily life, create chaos in business operations, and result in significant financial losses [8, 19, 39].

*WWW '25, April 2025, Sydney, Australia*
© 2024 Copyright held by the owner/author(s). Publication rights licensed to ACM.
ACM ISBN 978-1-4503-XXXX-X/18/06
https://doi.org/XXXXXXX.XXXXXXX

To ensure high reliability and availability of these systems, numerous AI-based methods [3–6, 13, 14, 16, 17, 22, 28, 29, 31, 32, 34, 36, 37, 42] have been proposed to detect anomalies from system operational data, such as service KPIs, as well as system runtime status data like CPU and memory usage. This data often takes the form of Multivariate Time Series (MTS). The objective of MTS anomaly detection is to determine whether time steps in the series are normal or abnormal.

Despite the vast amount of time series generated daily in these systems, supervised learning methods often face challenges in this domain due to the labor-intensive data labeling process and the scarcity of anomalous instances [14, 36]. As a result, MTS anomaly detection is typically framed as a semi-supervised learning task, where only normal data is available for model training.

Traditional machine learning methods, such as Local Outlier Factor (LOF) [4], One Class Support Vector Machine (OCSVM) [28], Isolation Forest (iForest) [16], have been widely used for anomaly detection tasks. These methods treat multivariate observations at time steps as points in a feature space. Distance or density metrics are used to assess the proximity of these points to each other. The points that deviate significantly from the majority are flagged as anomalies. We refer to this approach as proximity-based. Recently, proximity-based methods that combine deep representation learning, such as DeepSVDD [26] and DAGMM [42], have also emerged. However, this approach often struggles with high accuracy due to their inability to effectively capture dynamic intra-variate temporal dependencies and complex inter-variate relationships.

To address the aforementioned issues, many temporal-based and spatiotemporal-based methods have been developed [5, 14, 15, 29, 30, 36, 37]. For instance, THOC [29] employs a differentiable hierarchical clustering mechanism to integrate temporal features across various scales and resolutions for effective normal pattern learning. AT [36] models prior and series associations between time steps in series for temporal pattern learning. DCdetector [37] utilizes two patch-based attention networks for contrastive learning to capture temporal dependencies in the given time series. We classify these approaches as temporal-based methods since they primarily rely on temporal dependencies for anomaly detection. In contrast, InterFusion [15] leverages a hierarchical Variational Autoencoder framework to capture both intra-variate temporal and inter-variate dependencies. Memto [30] presents a memory-guided reconstruction approach that utilizes a single Transformer network to capture temporal dependencies, while also integrating inter-variate associations through the interactions between the derived representations and a set of memory items. STEN [5] presents a framework that combines subsequence order prediction, capturing temporal correlations, with distance prediction, which learns spatial relationships between sequences. These methods are categorized into spatiotemporal-based methods. While both temporal-based and spatiotemporal-based methods enhance the ability to capture

intricate intra-variate or/and inter-variate dependencies within time series, they often fail to capture salient intra-variate temporal and inter-variate dependencies in time series due to their focus on excessively fine granularity. This limitation can result in their suboptimal performance in MTS anomaly detection.

This paper presents MtsCID, a novel semi-supervised anomaly detection approach for **M**ultivariate **T**ime **S**eries through capturing **Coarse**-grained **I**ntra-variate and inter-variate **D**ependencies. MtsCID employs a dual-network architecture: one network utilizes the attention maps of multi-scale intra-variate patches to capture coarse-grained temporal dependencies between time steps, while the other focuses on variate interactions, leveraging convolutions, frequency component-based Transformer and a set of sinusoidal prototypes to capture coarse-grained inter-variate relationships. The deviation from normal patterns in each dimension is aggregated to generate losses during training and anomaly scores for each time step during inference. The resulting anomaly score indicates whether a given timestamp is anomalous or not. Our approach has been evaluated on seven commonly used publicly available datasets. Experimental results demonstrate its effectiveness, achieving comparable or superior anomaly detection performance compared to nine state-of-the-art methods.

In summary, our main contributions are as follows:

(1) **A novel time-frequency interleaved learning scheme:** We introduce an novel scheme for learning intra- and inter-variate dependencies through interleaved processing in both the time and frequency domains. This method utilizes frequency domain components to align inter-variate time steps and time domain representation to learn coarse-grained temporal dependencies, thereby enhancing the capture of normal patterns within the series and improving anomaly detection.

(2) **A dual-network multivariate time series anomaly detection approach:** We propose MtsCID, an anomaly detection method that utilizes a dual-network architecture for coarse-grained learning of both intra-variate temporal dependencies and inter-variate relationships. This design enriches the information embedded in the representations, enhancing the overall performance of time series anomaly detection.

(3) **Extensive experiments:** We compare MtsCID with nine SOTA baselines on seven widely used public datasets. The results confirm the effectiveness of MtsCID. In addition, The ablation experimental results show the efficacy of each major component in MtsCID.

## 2 Proposed Approach

### 2.1 Problem Definition

Given a set of subsequences $D = \{X^1, \ldots, X^N\}$, where $N$ represents the total number of subsequences and each $X^i \in \mathbb{R}^{T \times C}$ denotes a subsequence of observations $[x_1^i, \ldots, x_L^i]$, with $L$ indicating the length of the subsequence. Here, $x_t^i \in \mathbb{R}^C$ represents the multivariate observation vector at time $t$, with $C$ indicating the total number of variates. Semi-supervised time series anomaly detection aims to identify anomalies at the individual time step level within specified subsequences, assuming that the training subsequences consists solely of normal observations.

### 2.2 Approach Overview

When MTS subsequences are input into MtsCID, they are processed through two branches: the upper branch for learning temporal dependencies and the lower branch for learning inter-variate relationships. As shown in Fig 1, the two branches comprises three components: the temporal autoencoder network (t-AutoEcoder), the inter-variate dependency encoder network (i-Encoder), and the sinusoidal prototypes interaction module (p-i Module).

In the upper branch, each variate in the subsequence is initially transformed into its frequency components. The fc-Linear, a frequency component-based Linear layer, and fc-Transformer, a frequency component-based Transformer network, are employed to learn the dependencies of these frequency components. The derived representations are subsequently transformed back to the time domain and passed through a set of intra-variate ts-Attention (time-series Attention) networks to learn temporal dependencies from the attention maps of multi-scale patches. Finally, these representations are fed into the decoder to reconstruct input sequences.

In the lower branch, each subsequence is first processed in the time domain using a convolutional layer with a specific kernel size to capture local temporal dependencies in the variates. The output is then fed into the inter-variate fc-Transformer networks to learn inter-variate relationships in the frequency domain. The resulting representations are subsequently interacted with a set of sinusoidal prototypes in the p-i Module for inter-variate relationship pattern learning.

During training, the reconstruction loss between the input and the generated sequences from the upper branch is combined with the output from the lower branch to form a comprehensive loss that guides model training. During inference, the reconstruction loss from the upper branch interacts with the output from the lower branch to produce an anomaly score for each time step in the series, indicating whether a specific timestamp is anomalous.

Figure 2 provides a brief overview of how the relevant building blocks function. Next, we will elaborate on each major component of MtsCID.

### 2.3 Temporal AutoEncoder Network (*t-AutoEncoder*)

The temporal autoencoder network is designed to capture temporal dependencies between time steps within the series. When time series subsequences $X \in \mathbb{R}^{B \times L \times C}$ are input—where $B$ represents the batch size, $L$ is the subsequence length, and $C$ denotes the number of series—each series is first transformed into its frequency components $H \in \mathbb{R}^{B \times f \times C}$. Here, $H$ encompasses the real and imaginary parts of the frequency components, with $F$ representing the number of frequency components. This transformation is achieved using the Discrete Fourier Transform (DFT).

The real and imaginary parts of the derived frequency components are projected into distinct latent spaces in the fc-Linear network using respective learnable parameters $W^{(r)}$ and $W^{(i)} \in \mathbb{R}^{C \times d}$. Two independent networks for processing real and imaginary parts are also applied in the subsequent fc-Transformer networks. For

Figure 1: The overview of MtsCID.

clarity and brevity, we will omit the subscripts in the following descriptions.

Subsequently, the fc-Transformer network processes the previous output to generate representations, $\hat{H} \in \mathbb{R}^{B \times f \times d}$ for each frequency component. Notably, the $Q$, $K$, and $V$ inputs to the fc-Transformer network are derived from the same source. In line with standard Transformer architecture, a residual connection and layer normalization are utilized to enhance these representations. The resulting representations are then transformed back to the time domain using the inverse Discrete Fourier Transform (iDFT) as $Z \in \mathbb{R}^{B \times L \times d}$.

Next, $Z$, generated from the previous step, is transformed into differently sized patches in a channel-independent manner, denoted as $\{Z^{p_1}, \ldots, Z^{p_m}\}$, where $Z^{p_i} \in \mathbb{R}^{(B \times d) \times n_i \times p_i}$ represents a matrix with patch number $n_i$ and patch size $p_i$, for $i \in \{1, \ldots, m\}$, with $m$ indicating the number of multi-scale patches. Each $Z^{p_i}$ is then fed into an intra-variate ts-Attention network to generate corresponding attention maps $A^{p_i} \in \mathbb{R}^{(B \times d) \times n_i \times n_i}$ for the patched subsequences, which capture intra-variate temporal dependencies at a specific granularity. The derived attention maps $A^{p_i}$ are subsequently mapped back to their original patch sizes using learnable parameters $M^{p_i} \in \mathbb{R}^{n_i \times p_i}$. All the projected attention maps are then averaged and transformed back into the input subsequence format through an unpatch operation, resulting in $\hat{Z} \in \mathbb{R}^{B \times L \times d}$. Finally, these $\hat{Z}$ are passed into the decoder module, consisting of a linear layer, to obtain the reconstructed sequences $\hat{X} \in \mathbb{R}^{B \times L \times C}$. The mathematical formulas for our Temporal AutoEncoder are as follows:

$$H = \mathrm{DFT}(X) \tag{1}$$

$$Q = K = V = HW \tag{2}$$

$$\hat{H} = \mathrm{Softmax}\left(\frac{QK^T}{\sqrt{d}}\right)V \tag{3}$$

$$Z = \mathrm{LayerNorm}\left(\mathrm{iDFT}(\hat{H} + V)\right) \tag{4}$$

$$Z^{p_i} = \mathrm{Patch}(Z) \tag{5}$$

$$A^{p_i} = \mathrm{Softmax}\left(\frac{Z^{p_i} Z^{p_i T}}{\sqrt{p_i}}\right) M^{p_i} \tag{6}$$

$$\hat{Z} = \mathrm{Unpatch}\left(\frac{1}{m}\sum_{i=1}^{m} A^{p_i}\right) \tag{7}$$

$$\hat{X} = \mathrm{Decoder}(\hat{Z}) \tag{8}$$

We opt for fc-Linear and fc-Transformer operating in the frequency domain based on two key assumptions: 1) Time series data in the frequency domain may reveal more salient patterns compared to those in the time domain, as they are less influenced by individual time steps. 2) Since time series values are continuous, their representations in the frequency domain significantly reduces their population. This reduction may enhance the determinism of subsequent reconstructions when learning the pattern from normal samples. In the later ablation study section, our experimental results will show that the effectiveness in detection performance using frequency domain representations is better than using time domain representations.

## 2.4 Inter-variate Dependency Encoder Network (*i-Encoder*)

Prior studies have demonstrated that inter-variate relationships can enhance anomaly detection in multivariate time series [5, 15, 30]. In this study, we employ an independent inter-variate dependency encoder network to capture these relationships from normal time series subsequences. Since each variate measures different aspects of the monitored system, they often exhibit varying periodicities, making it challenging to learn their normal combination patterns at time steps.

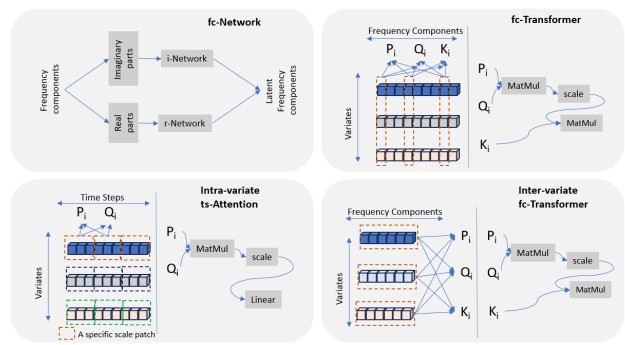

**Figure 2: The building blocks in MtsCID.**

To address this challenge, we first input the time series subsequences $X \in \mathbb{R}^{B \times L \times C}$ into a 1D-convolution network. The kernel size of the convolution network is assumed as $k$. This process yields corresponding representations, $T \in \mathbb{R}^{B \times C \times L}$, which capture local temporal dependencies in variates. The derived representation offers two key benefits: 1) Capturing coarse-grained temporal dependencies enhances the semantic representations of individual time steps, making it more robust to noise interference. 2) It can mitigate the issues related to potential misalignment of time steps across variates that may occur during data collection, as well as challenges posed by unsynchronized variates.

The derived representations $T$ is then transformed into its frequency components, $E \in \mathbb{R}^{B \times C \times f}$. Then, a fc-Transformer network is employed to capture inter-variate relationships in the frequency domain, generating the representations $J \in \mathbb{R}^{B \times C \times f}$. Notably, the inter-variate fc-Transformer processes data along the variate dimension, while the intra-variate fc-Transformer in the temporal autoencoder operates along the time step dimension. A detailed comparison can be seen in Figure 2.

The derived representations $J$, which capture inter-variate relationships, are then transformed back into the time domain. A residual connection and layer normalization are applied, resulting in the final representations $O \in \mathbb{R}^{B \times L \times C}$. The mathematical expressions summarizing this process are presented as follows:

$$T = Conv1d(X) \tag{9}$$

$$E = DFT(T) \tag{10}$$

$$J = Softmax(\frac{EE^T}{\sqrt{f}})E \tag{11}$$

$$O = LayerNorm(iDFT(J) + X) \tag{12}$$

## 2.5 Sinusoidal Prototypes Interaction Module (*p-i Module*)

In the inter-variate dependency encoder network, while the derived representations $O$ capture inter-variate relationships, the combinations of variables at time steps remain complex and challenging for learning salient normal patterns. To address this issue, we aim to simplify these complex combinations into a limited set, making it easier to learn normal inter-variate relationship patterns.

In the study by Song et al. [30], the authors demonstrate that incorporating memory items, also known as prototypes, can enhance the learning of inter-variate normal patterns. Inspired by their work, we develop a sinusoidal Prototypes Interaction Module. Unlike their dynamic memory updating mechanism, we utilize a set of fixed memory items $M \in \mathbb{R}^{C \times L}$, derived from sinusoidal functions with varying periodicity, defined as follows:

$$M \in \mathbb{R}^{L \times C}, \quad M_{i,j} = \cos\left(\frac{2\pi}{L} \cdot i \cdot j\right)$$

$$\text{for } i = 0, 1, \dots, L-1 \text{ and } j = 0, 1, \dots, C-1.$$

By using fixed memory items, we avoid the instability issue in model training mentioned in [30]. Furthermore, since these memory items are derived from sinusoidal functions with different periodicity, their combinations along the time step dimension approximate a limited set. As shown in the experimental section, this approach enhances the salience of patterns across inter-variate relationships, thereby improving both robustness and detection accuracy, even without the need for additional two-phase training and clustering processes as described in [30].

Following the practice in [30], we multiply representations $O$, generated from the inter-variate dependency encoder network, with our fixed sinusoidal prototypes through dot product, followed by the SoftMax operation, as follows:

$$w_{ti} = \frac{\exp(< O_{:,t,:}, M_{i,:} > /\tau)}{\sum_{j=1}^{L} \exp(< O_{:,t,:}, M_{j,:} > /\tau)} \tag{13}$$

## 2.6 Learning Tasks

MtsCID utilizes two learning tasks, i.e., temporal dependency reconstruction task and prototype-oriented learning task, to effectively guide model optimization during training. These tasks corresponds to two specific losses: the $L_{t-rec}$, and the $L_{i-ent}$.

*2.6.1 Temporal Dependency Reconstruction Task.* For the temporal dependency learning branch, i.e. the upper branch, a reconstruction loss is utilized between the input and the reconstructed ones to direct its network for optimization. The reconstruction loss $L_{t-rec}$ is defined as $L2$ loss between $X$ and $\hat{X}$:

$$L_{t-rec} = \frac{1}{B} \sum_{s=1}^{B} \|X^s - \hat{X}^s\|_2^2 \tag{14}$$

*2.6.2 Prototype-Oriented Learning Task.* For the inter-variate dependency learning branch, i.e. the lower branch, we adopt the practice from the study [30] that an entropy loss $L_{i-ent}$ as our auxiliary loss for regularization on $W$ derived in Equation 13:

$$L_{i-ent} = \frac{1}{B} \sum_{s=1}^{B} \sum_{t=1}^{L} \sum_{i=1}^{C} -w_{t,i} log(w_{t,i}) \tag{15}$$

During the training phase, The objective function $L$ is to minimize the combination of loss term Equation (14) and Equation (15) as follows:

$$L = L_{t-rec} + \lambda L_{i-ent} \tag{16}$$

where $\lambda$ denotes a hyper-parameter for weighting coefficient.

**Table 1: Overview of datasets used in the experiments.**

| Datasets | #Entities | #Dims. | Training #Timesteps | Testing #Timesteps | Testing %Anomalies |
|---|---|---|---|---|---|
| SMAP | 55 | 25 | 135,183 | 427,617 | 13.13% |
| MSL | 27 | 55 | 58,317 | 73,729 | 10.72% |
| SMD | 28 | 25 | 708,405 | 708,420 | 4.16% |
| PSM | 1 | 25 | 55,541 | 34,387 | 3.13% |
| SWaT | 1 | 51 | 496,800 | 449,919 | 11.98% |
| GECCO | 1 | 9 | 69,260 | 69,260 | 1.05% |
| SWAN | 5 | 38 | 60,000 | 60,000 | 32.6% |

## 2.7 Anomaly Detection

During inference, the deviations from the learned normal patterns in the two branches, specifically the Temporal Deviation ($TD$) and Relationship Deviation ($RD$), are combined to generate anomaly scores for each time step in the input. The $TD(X_{t,:}, \hat{X}_{t,:})$ is defined as the distance between the input $X_{t,:} \in \mathbb{R}^C$ and the reconstructed input $\hat{X}_{t,:} \in \mathbb{R}^C$ at time $t$. The $RD(O_{t,:}, M_{:,:})$ is defined as the distance between each $O_{t,:}$ and its nearest memory step $m_{s,:}$. The formal definitions of the Anomaly Scores (AScore) are as follows:

$$AScore(X) = Softmax([RD(O_{t,:}, M_{:,:})]) \circ [TD(X_{t,:}, \hat{X}_{t,:})] \quad (17)$$

where $\circ$ is element-wise multiplication and and $AScore(X) \in \mathbb{R}^L$ is anomaly score at each time step. These anomaly scores with higher scores indicating a higher likelihood of the corresponding time steps as anomalies.

## 3 Experimental Setting

### 3.1 Datasets

We evaluate MtsCID on two groups of seven widely used real-world MTS datasets. A summary of these datasets is provided in Table 1, along with further descriptions below.

- SMAP (Soil Moisture Active Passive) [11] is a dataset of soil samples and telemetry information using the Mars rover by NASA, while MSL (Mars Science Laboratory) [11] corresponds to the sensor and actuator data for the Mars rover itself. SMD (Server Machine Dataset) [31] is a five-week long dataset of stacked traces of the resource utilization of 28 machines from a compute cluster. PSM (Pooled Server Metrics) [1] is collected internally from multiple application server nodes at eBay with 25 dimensions. SWaT (Secure Water Treatment) [20] is gathered from a real-world water treatment plant, including 7 days of normal operations and 4 days of abnormal operations.
- NIPS-TS-GECCO [21], referred to as GECCO, comprises drinking water quality data for the Internet of Things and was published at the 2018 Genetic and Evolutionary Computation Conference. NIPS-TS-SWAN [2, 12], known as SWAN, is an openly accessible, comprehensive MTS benchmark derived from solar photospheric vector magnetograms in the Space Weather HMI Active Region Patch series. Both datasets are sourced from [37].

## 3.2 Evaluation Metrics

In this study, we employ three groups of evaluation metrics.

- The first group of metrics includes point-adjustment Precision, Recall, and F1-score, which are widely used in time series anomaly detection [30, 31, 34–38, 40]. This approach acknowledges that anomalies typically manifest as contiguous segments rather than isolated points. Consequently, if any point within a contiguous segment is detected as anomalous, the entire segment is deemed correctly identified. Following the methodology in [5], we utilize the best F1 score to mitigate biases from threshold settings.
- The second group is Affiliation Precision, Recall and F1 Score [10], which are also used in recent studies [5, 22, 37]: This set of metrics incorporates duration measures between ground truth and predictions, addressing limitations of other metrics that ignore temporal adjacency and event duration. Due to space constraints, we present only the Affiliation F1-score (AF-F1).
- VUS-ROC/VUS-PR [23] are used as the third group of metrics in our study, which are also used in recent studies [5, 22, 37]. These metrics extend the ROC-AUC and PR-AUC measures. VUS-ROC (Volume Under the ROC Surface) and VUS-PR (Volume Under the PR Surface) address biases introduced by point adjustment by evaluating the overall volume under the respective curves.

## 3.3 Implementation and Environment

Following the approach in [30], we generate sub-sequences using a non-overlapping sliding window of length 100 to create fixed-length inputs for each dataset. The training data is then divided into 80% for training and 20% for validation. In the t-AutoEncoder network, we set the number of multi-scale patches to $m = 2$, with $p_i$ taking values from [10, 20]. The i-Encoder network is configured with a 1D convolution layer featuring a kernel size of ($k = 5$).

We implemented MtsCID using PyTorch 1.11.0. The model was trained with the AdamW optimizer and employed polynomial learning rate decay, starting at $2 \times 10^{-3}$ and gradually decreasing to $5 \times 10^{-5}$. A mini-batch size of 64 was used, and training continued for up to 20 epochs, with early stopping applied if performance did not improve for 10 consecutive iterations. All experiments were conducted on a Linux server Ubuntu 20.04 equipped with an AMD Ryzen 3.5GHz CPU, 96 GB of memory, and an RTX2080Ti with 11 GB of GPU memory.

## 4 Results and Analysis

### 4.1 The Effectiveness of MtsCID

To evaluate the effectiveness of our proposed method, we compare MtsCID with nine state-of-the-art semi-supervised methods. The baseline methods include proximity-based approaches—iForest [16], DeepSVDD [26], and DAGMM [42]; temporal-based methods—THOC [29], Anomaly Transformer [36], and DCdetector [37]; spatio-temporal-based models—InterFusion [15], MEMTO [30], and STEN [5]. In our comparative analysis, the implementations of the baseline approaches were obtained from their public repositories. To ensure

**Table 2: Overall Performance Comparison of All Methods in Point-Adjustment Metrics.**

| Method | SMD | | | MSL | | | SMAP | | | SWaT | | | PSM | | |
|---|---|---|---|---|---|---|---|---|---|---|---|---|---|---|---|
| | P | R | F1 | P | R | F1 | P | R | F1 | P | R | F1 | P | R | F1 |
| iForest[16] | 42.31 | 73.29 | 53.64 | 53.94 | 86.54 | 66.45 | 52.39 | 59.07 | 55.53 | 49.29 | 44.95 | 47.02 | 76.09 | 92.45 | 83.48 |
| DeepSVDD[26] | 78.54 | 79.67 | 79.10 | 91.92 | 76.63 | 83.58 | 89.93 | 56.02 | 69.04 | 80.42 | 84.45 | 82.39 | 95.41 | 86.49 | 90.73 |
| DAGMM [42] | 67.30 | 49.89 | 57.30 | 89.60 | 63.93 | 74.62 | 86.45 | 56.73 | 68.51 | 89.92 | 57.84 | 70.40 | 93.49 | 70.03 | 80.08 |
| THOC [29] | 79.76 | 90.95 | 84.99 | 88.45 | 90.97 | 89.69 | 92.06 | 89.34 | 90.68 | 83.94 | 86.36 | 85.13 | 88.14 | 90.99 | 89.54 |
| A.T [36] | 91.33 | 94.50 | 92.88 | 92.09 | 96.23 | 94.10 | 94.32 | 99.02 | 96.61 | 92.00 | 99.08 | 95.40 | 98.08 | 98.31 | 98.19 |
| DCdetector [37] | 84.14 | 88.60 | 86.28 | 90.42 | 94.14 | 92.21 | 95.32 | 97.86 | 96.57 | **96.87** | 99.21 | **98.02** | 98.21 | 98.27 | 98.24 |
| InterFusion [15] | 87.02 | 85.43 | 86.22 | 81.28 | 92.70 | 86.62 | 89.77 | 88.52 | 89.14 | 80.59 | 85.58 | 83.01 | 83.61 | 83.45 | 83.52 |
| MEMTO [30] | 89.17 | 94.68 | 91.84 | 92.25 | 96.10 | 94.13 | 93.80 | **99.41** | 96.52 | 92.83 | **99.96** | 96.26 | 98.37 | **99.04** | 98.50 |
| STEN [5] | 83.58 | 83.11 | 83.29 | 90.17 | 94.91 | 92.42 | **96.49** | 96.52 | 96.49 | 91.83 | 99.12 | 95.31 | 97.74 | 98.02 | 97.88 |
| **MtsCID** | **91.50** | **95.37** | **93.39** | **93.37** | **96.97** | **95.13** | 95.90 | 98.79 | **97.32** | 94.16 | 99.82 | 96.91 | **98.57** | 98.51 | **98.54** |

[1] P, R, and F1 refer to Precision, Recall, and F1-score, respectively. The results represent percentages.

[2] We reproduced the results for Anomaly Transformer, DCdetector, MEMTO, and STEN, while adopting the reported performance from [30] for the other baselines.

[3] In the table, values that are underlined represent the second-best metrics, while those in bold indicate the best metrics.

**Table 3: Multi-Metrics Performance Comparison Results on Recent SOTA Methods.**

| Method | Metrics | SMD | MSL | SMAP | SWaT | PSM | GECCO | SWAN |
|---|---|---|---|---|---|---|---|---|
| **Anomaly Transformer [36]** | F1 | 92.88 | 94.10 | 96.61 | 95.40 | 98.19 | 44.53 | 73.86 |
| | AF-F1 | 74.11 | 67.54 | 67.31 | 53.22 | 65.90 | 70.37 | 7.30 |
| | VUS-PR | 72.53 | 84.83 | 92.18 | 95.06 | 92.48 | 10.14 | 90.99 |
| | VUS-ROC | 82.89 | 94.33 | 97.66 | 98.39 | 94.18 | 61.66 | 89.38 |
| **DCdetector [37]** | F1 | 86.28 | 92.21 | 96.57 | **98.02** | 98.24 | 37.08 | 73.59 |
| | AF-F1 | 66.04 | 66.91 | 67.68 | 69.75 | 63.78 | 63.19 | 6.02 |
| | VUS-PR | 60.79 | 83.40 | 92.39 | **97.32** | 91.10 | 10.08 | 91.83 |
| | VUS-ROC | 78.63 | 94.45 | 97.32 | **98.90** | 90.54 | 60.19 | 88.55 |
| **MEMTO [30]** | F1 | 91.84 | 94.13 | 96.52 | 96.26 | 98.50 | 54.25 | 73.93 |
| | AF-F1 | 70.71 | 67.27 | 66.72 | 34.97 | 66.46 | 16.21 | 0.53 |
| | VUS-PR | 72.58 | 85.93 | 92.17 | 95.58 | 94.00 | 17.96 | **93.68** |
| | VUS-ROC | 82.38 | 88.87 | 97.09 | 98.43 | 92.72 | 61.98 | 86.33 |
| **STEN [5]** | F1 | 83.29 | 92.42 | 96.49 | 95.31 | 97.88 | 36.34 | 73.85 |
| | AF-F1 | 64.02 | 63.46 | 66.86 | **70.98** | 59.94 | 48.44 | 2.65 |
| | VUS-PR | 61.37 | 85.26 | **94.10** | 90.62 | 94.70 | 15.74 | 92.92 |
| | VUS-ROC | **91.29** | 95.59 | 98.30 | 98.38 | 96.71 | 86.06 | 92.11 |
| **MtsCID** | F1 | **93.39** | **95.13** | **97.32** | 96.91 | **98.54** | **77.10** | **74.29** |
| | AF-F1 | **74.46** | **68.20** | 67.68 | 57.01 | 67.56 | **73.40** | 8.69 |
| | VUS-PR | **79.12** | **87.61** | 94.02 | 95.83 | 93.50 | **35.90** | 93.62 |
| | VUS-ROC | 84.22 | 89.46 | 96.71 | 98.30 | 91.59 | 72.38 | 86.45 |

[1] Underlined figures represent the second-best metrics, while those in bold indicate the best metrics.

consistency, we adhered to the parameters provided by their respective implementations unless otherwise specified. Each method was executed five times for each dataset, and the resulting values were averaged to report the final results.

We first evaluate MtsCID against all the aforementioned baselines using point-adjustment metrics across five datasets, with results presented in Table 2. Since recent methods, including Anomaly Transformer, DCdetector, MEMTO, and STEN, outperform other baselines, we focus our multi-metric comparison of MtsCID primarily on these models. This comparison includes the aforementioned

five datasets, as well as two additional, more challenging datasets (GECCO and SWAN) that feature a wider variety of anomalies.

The experimental results in Tables 2 and 3 demonstrate that MtsCID achieves robust and superior performance across all datasets. Specifically, MtsCID secures the highest F1 and AF-F1 scores in six out of seven datasets, and the second-best F1 score in the remaining dataset, outperforming all baseline methods. Notably, on the challenging GECCO dataset, MtsCID demonstrates a significant advantage in the F1 metric, outperforming the second-best baseline

by 42.12%. MEMTO also achieves consistently excellent results, indicating that leveraging prototypes enhances detection performance by providing additional information. While DCdetector and STEN perform well, closely matching MtsCID's effectiveness across most datasets, there is a notable disparity in detection effectiveness on the SMD and GECCO dataset.

Furthermore, We can see from Table 2 and Table 3 that the methods leveraging temporal and spatiotemporal dependencies consistently outperform all proximity-based methods, underscoring the importance of capturing dependencies among features in time series. However, spatiotemporal-based methods like STEN do not always surpass temporal-based methods such as Anomaly Transformer and DCdetector, likely because temporal dependencies are the primary indicators for identifying anomalies. This suggests that spatial features require careful design for improved performance.

## 4.2 Ablation studies

In this section, we aim to thoroughly examine the effectiveness of each major component within MtsCID on the final results. To accomplish this, we conduct ablation studies that categorize the MtsCID variants into three distinct groups:

- **Network Branch Ablation:** We compare variants that exclude either the upper branch or lower branch.
- **Coarse-Grained Processing Ablation:** We compare variants that either exclude the intra-variate ts-Attention layer in the t-AutoEncoder network or replace the convolution layer with a linear layer in the i-Encoder network.
- **Frequency Processing Ablation:** We evaluate variants that replace all frequency component-based networks with time domain counterparts.

The results are presented in Table 4. It is clear that MtsCID with dual networks (without subscripts) consistently outperforms single network counterparts (where "to" indicates the t-AutoEncoder branch and "io" indicates the i-Encoder branch). These findings empirically support our hypothesis that integrating both temporal dependency and inter-variate relationship learning in MtsCID enhances the model's ability to learn patterns from normal time series, facilitating better anomaly detection.

The results in Table 4. also show that MtsCID with coarse-grained processing consistently outperforms their counterparts without one of the coarse-grained treatments. This empirically confirms that coarse-grained processing in temporal dependency learning and inter-variate relationship learning in MtsCID enhances the model's ability to learn patterns from normal time series, facilitating better anomaly detection.

Table 4 also provides a clear comparison between MtsCID and its variants subscripts with $td$ (The t-AutoEncoder and i-Encoder both operate on the time domain rather than on the frequency domain) across datasets. The experimental results empirically support our claim that working on the frequency domain facilitates the trained model in effectively learning normal patterns.

## 4.3 Sensitivity studies

*4.3.1 Loss Weights Sensitivity.* In the previous experiments, all assessments were conducted with the hyperparameter set at $\lambda = 0.1$. To further investigate the impact of this weight, we performed a sensitivity analysis by varying the hyperparameter within the range of $10^{-3}$ to $10^2$. As shown in Figure 3a, the model's performance is largely insensitive to the variations in hyperparameter choices for the loss weight. Therefore, we have decided to keep the current hyperparameter setting.

*4.3.2 Sensitivity Analysis of Multi-Scale Patch Settings.* To explore the impact of multi-scale patch settings, we conducted a sensitivity analysis with patches configured as $\{[5, 10], [10, 20], [5, 10, 20]\}$. As illustrated in Figure 3b, detection performance showed slight variations based on patch settings. Consequently, we have chosen to use the $[10, 20]$ configuration.

*4.3.3 Sensitivity Analysis of Convolution Kernel Settings.* To further investigate the impact of kernel settings, we conducted a sensitivity analysis with kernels set to $\{1, 3, 5, 7, 9\}$. As shown in Figure 3c, performance slightly fluctuated with different kernel settings, except for the SMD dataset, where performance decreased as kernel size increased. A kernel size of 5 yielded relatively higher results across the other datasets, leading us to choose this size.

## 4.4 Scalability Studies

To evaluate the scalability of MtsCID, we conducted runtime comparisons with baseline methods, highlighting the efficiency of our proposed approach. Figure 3d presents the average training and testing times per epoch for these methods. For conciseness, we focus on the experimental results from the MSL dataset, where MtsCID demonstrates superior efficiency in both training and testing compared to all the other baseline methods. This indicates that MtsCID has strong scalability potential for real-world applications.

## 5 Discussion

### 5.1 Why does MtsCID Work?

In this section, we present two key reasons why MtsCID outperforms relevant baseline methods. First, MtsCID extracts salient patterns from attention maps generated by multi-scale patches and inter-variate relationships revealed through convolution operations, rather than relying on excessively fine-grained time steps, resulting in improved performance. Second, by integrating frequency domain processing with time domain operations, MtsCID aligns time steps across variates, reducing interference from misalignment. This combination enhances its ability to capture normal patterns from inter-variate relationships, ultimately improving detection performance.

### 5.2 Limitations and Future Work

While our experiments demonstrate the effectiveness of MtsCID for MTS anomaly detection, it does have limitations. Currently, the temporal dependencies between time steps and inter-variate relationships are learned independently during training, which may lead to the loss of valuable information that could enhance anomaly identification. In future work, we plan to explore self-supervised techniques to better capture these connections and improve performance. Additionally, the utilization of information in the frequency domain remains underdeveloped, making it worthwhile to investigate this further in our upcoming research.

Table 4: Ablation Experiments for Network Architecture and Operating Domain.

| Category | Ablation Component | Method Invariant | SMD F1 | MSL F1 | SMAP F1 | SWaT F1 | PSM F1 |
|---|---|---|---|---|---|---|---|
| Network | **i-Encoder Network** | MtsCID$_{to}$ | 88.45 | 91.96 | 95.18 | 91.45 | 96.58 |
| | **t-AutoEncoder Network** | MtsCID$_{io}$ | 83.23 | 89.84 | 94.09 | 96.42 | 97.02 |
| Granularity Processing | **Multi-Scale Patch Attention** | MtsCID$_{co}$ | 93.13 | 94.91 | 96.87 | 96.78 | 98.41 |
| | **Convolution** | MtsCID$_{ao}$ | 92.05 | 94.73 | 97.01 | 96.41 | 98.11 |
| Domain | **Frequency Domain Processing** | MtsCID$_{td}$ | 91.22 | 94.89 | 96.72 | 96.65 | 98.43 |
| / | **With All Components** | MtsCID | **93.39** | **95.13** | **97.32** | **96.91** | **98.54** |

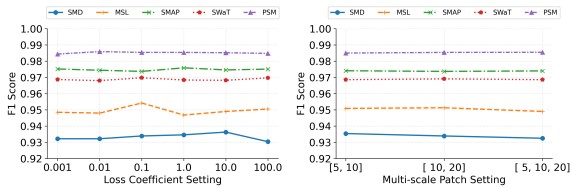

(a) Loss coefficient sensitivity. (b) Patch setting sensitivity.

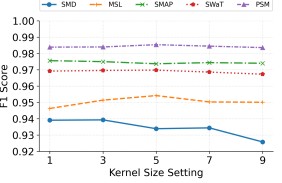

(c) Kernel setting sensitivity.

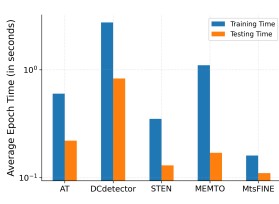

(d) Scalability analysis.

Figure 3: Sensitivity and Scalability Analysis.

## 6 Related Work

*Existing MTS Anomaly Detection Methods.* From the perspective of label utilization, existing methods can be categorized into supervised, semi-supervised, and unsupervised approaches. Supervised methods, such as AutoEncoder [27], LSTM-VAE [24], Spectral Residual [25], and RobustTAD [9], deliver competitive performance but are limited by the labor-intensive labeling process and the scarcity of anomalies. Unsupervised methods, like GANF [7] and MTGFlow [41], eliminate the need for labels but often produce sub-optimal results due to the lack of guidance. In contrast, semi-supervised methods—including all baseline methods in our experimental evaluation, such as THOC [29], InterFusion [15], Anomaly Transformer [36], DCdetector [37], MEMTO [30], and STEN [5]—leverage abundant normal data while reducing reliance on rare anomalies. This approach enhances detection performance and eases data requirements. Our proposed method also follows a semi-supervised approach.

From the perspective of feature utilization, existing methods for MTS anomaly detection can be categorized into step-based and attention map-based approaches. Step-based methods learn normal patterns from individual time steps in variates or their latent representations, with notable examples including iForest [16], DAGMM [42], DeepSVDD [26], THOC [29], InterFusion [15], MEMTO [30], and STEN [5]. Since each time step plays a significant role in these methods, they often struggle to capture meaningful semantics in variates effectively. In contrast, attention map-based methods leverage attention mechanisms to analyze relationships across segments of time steps. By focusing on attention maps rather than each time step, these methods capture more salient patterns, leading to improved results. Representative approaches include the Anomaly Transformer [36] and DCdetector [37]. In our proposed method, we utilize attention maps as features in one branch of the framework to facilitate intra-variate temporal dependency learning.

*Prototype-based Time Series Anomaly Detection.* Prototype-based approach leverages a set of prototypes, representing normal patterns extracted during training, to improve anomaly differentiation. A notable example is MEMTO [30], which leverages prototypes constructed by a two-phase training process that combines clustering with incremental updates to enhance anomaly identification. In our proposed method, we employ a set of fixed prototypes that do not require updates. This approach mitigates instability during model training and eliminates additional processing like clustering and two-phase training, enhancing both robustness and time efficiency while maintaining comparable performance.

## 7 Conclusion

In this paper, we introduce MtsCID, a novel semi-supervised approach to MTS anomaly detection. MtsCID features a dual-network framework: an intra-variate temporal dependency learning network for capturing coarse-grained temporal patterns from attention maps, and an inter-variate relationship learning network combined with a sinusoidal prototypes interaction module for inter-variate relationship learning. This design is further enhanced by networks operating in the frequency domain, allowing the model to leverage information from both the time and frequency domains for effective normal pattern learning. Through the collaboration of these components, MtsCID effectively captures highly discriminative normal patterns from temporal dependencies and inter-variate relationships, enabling better discrimination of anomalous timestamps in the time series. Our extensive experiments on seven widely used public datasets demonstrate the effectiveness of MtsCID.

Our source code and experimental data are available at **https://anonymous.4open.science/r/MtsCID-2D2B/**.

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
