# OpenReview forum: "Multivariate Time Series Anomaly Detection by Capturing Coarse-Grained Intra- and Inter-Variate Dependencies"
_ACM.org/TheWebConf/2025/Conference — WWW 2025 Oral_

### Official Review · Reviewer_dbmn · 2024-11-14

**Novelty:** 6
**Technical Quality:** 7

**Review:**

**Summary:**

The paper proposes a method that utilizes multi-scale patching, 1D convolution, time-frequency interleaved learning, Transformer-like modules, along with two tasks, for semi-supervised multivariate time series anomaly detection and achieves results comparable or superior to state-of-the-art methods.

**Quality:**

The paper is of great quality in terms of writing and overall elaboration.

**Clarity:**

The paper is clear in both the method and experiments.

**Originality:**

The paper is original as a whole but not original in terms of the individual modules it used.

**Significance:**

The paper brings some insights into the research field.

**Pros:**

1. The paper is well-written, easy to understand, the diagrams and figures are well shown.

2. The author successfully proposed a method that is comparable to and even outperforms existing methods, with elaborate experiments to support it.

**Cons:**

1. For me, this paper basically stitches up several methods and combines all sorts of modules together. Considering all the modules it used, there should be way more possible combinations of modules, so the ablation study does not seem sufficient. Plus, according to the ablation study in the paper, the differences between the F1 scores of different variants are not numerically significant, it’s not very convincing.

2. It is unclear why the authors claim that their “coarse-grained” method is better than existing methods with “excessively fine granularity”, the explanation provided is insufficient, other than doing this can enhance the semantic representations and make it robust to noises. The modules related to “coarse-grained” processing, such as multi-scale patching and 1D convolution, are commonly used by researchers.

3. Also there are other important modules in the paper that are not implied in the title, which are inter-variate attention and frequency processing. “coarse-grained” only indicates multi-scale patching and 1D convolution. The major contribution and insight of the paper should not be “coarse-grained is better than excessively fine-grained”, it should be “combining multi-scale patching, 1D convolution, frequency processing, inter- and inter-variate attention, reconstruction and prototype learning, produces great results”.

**Minors:**

Line 557, [10, 20] looks like a citation, it would be better to change the brackets to a different notation.

**Questions:**

1. Why is the upper branch of the model used for reconstruction and the lower branch for prototype-oriented learning? What is the reason for this, and can they be reversed?

2. Why is the anomaly score obtained through multiplication rather than addition?

3. For the upper branch can the data be processed along the time dimension first and then along the frequency dimension?

**Reviewer Confidence:**

3: The reviewer is confident but not certain that the evaluation is correct

**Scope:**

3: The work is somewhat relevant to the Web and to the track, and is of narrow interest to a sub-community

---

### Official Review · Reviewer_dLLQ · 2024-11-29

**Novelty:** 4
**Technical Quality:** 4

**Review:**

This paper proposes a two-branch anomaly detection framework for time series unsupervised anomaly detection. In both branches, this work deals the time series both in the frequency domain and time domain. In the first branch, it uses ts-attention to extract intra-variate time series information. In the second branch, it uses cnn and fc-transformer to extract inter-variate time series information. Besides, it uses a stable memory items to improve the method performance.

### Strength

1. Overall well written. The paper is easy to follow and understand.
2. The authors have made extensive experiments to prove the effective of their methods. (including 7 datasets, 10 baselines and 6 metrics).

### Weakness
1. This paper lacks a clear main thread (the motivation for combining various modules together is not clear and lacks of a main challenge to be solved). It gives the impression that whatever modules are popular or useful recently are simply integrated into the current framework for use, and the framework is not designed to target at a main problem.
2. The irrationality of stable memory items. The standard prototypes are extracted from historical data, but in this paper the prototypes are set manually by the authors. When the prototypes are set manually, how could it represent the patterns of the dataset?
2. Some symbols are repeatedly set. For example, in line 168, "$X^i \in R^{T\times C}$", the length of subsequence is denoted by $T$, but in line 169, the authors use another symbol $L$ to denote it.

**Questions:**

Please refer to the weakness.

**Reviewer Confidence:**

3: The reviewer is confident but not certain that the evaluation is correct

**Scope:**

3: The work is somewhat relevant to the Web and to the track, and is of narrow interest to a sub-community

---

### Official Review · Reviewer_BFtj · 2024-12-01

**Novelty:** 6
**Technical Quality:** 6

**Review:**

This paper introduces MtsCID, a novel semi-supervised approach for Multi-Time Series (MTS) anomaly detection. The proposed method integrates a dual-network framework that captures both intra-variate temporal dependencies and inter-variate relationships through attention mechanisms and convolutional operations. Additionally, the use of frequency domain processing enhances the model's ability to detect anomalies in both time and frequency domains. The extensive experimental evaluation on multiple datasets demonstrates that MtsCID outperforms several baseline methods, particularly in its ability to capture discriminative normal patterns and detect anomalies effectively.

###Pros
1. The combination of attention maps, convolutional operations, and frequency domain processing is novel and contributes significantly to the advancement of MTS anomaly detection.
2. The authors provide an extensive set of ablation experiments, sensitivity analysis, and comparisons to existing methods, clearly demonstrating the effectiveness of MtsCID across various datasets.
3. The paper is well-organized, with a coherent flow from motivation to methodology, experiments, and conclusions.
4. The method’s ability to detect anomalies in time series data has broad applications in fields such as sensor monitoring, financial forecasting, and healthcare.

###Cons
1. While the paper mentions that MtsCID leverages both frequency-domain transformations and attention mechanisms, it lacks an in-depth discussion on how these operations scale with larger datasets. Specifically, the computational burden of these mechanisms, especially as the number of variates or time steps increases, remains unclear. Given the increasing size of real-world datasets, a discussion on how the method performs with larger datasets (e.g., longer time series or higher-dimensional data) would be beneficial.
2. The paper includes some time comparisons between MtsCID and other methods but does not sufficiently analyze the computational complexity of the model. How does the model’s time complexity grow with dataset size? Does it become computationally prohibitive as the dataset grows? Understanding the model's efficiency is critical for practical applications, especially when scaling to large, high-dimensional datasets.
3. While the paper evaluates MtsCID on multiple public datasets, the representativeness and diversity of these datasets are not fully discussed. Are these datasets sufficient to cover all possible types of anomalies and real-world scenarios? The paper should provide more context on the dataset characteristics, including class imbalances, data distribution, anomaly types (e.g., spikes, missing values, noise), and temporal dependencies. Additionally, the effect of different time spans and sampling frequencies across datasets (e.g., SWaT vs. SMD) should be explored, as these factors could influence model performance.
4. The paper does not discuss the practical limitations of MtsCID in real-world applications. For example, in industrial monitoring or financial risk prediction, time series data may contain noise, missing values, or other irregularities. The paper should address how MtsCID handles these challenges and whether the model is robust enough to work in practical settings where data quality can be suboptimal.

**Questions:**

1. How does MtsCID scale with larger datasets? Specifically, what is the impact on computational time and memory usage as the dataset size (in terms of time steps and variates) increases? Are there any optimization techniques that could be used to reduce the computational burden in large-scale applications?
2. Can you provide a more detailed analysis of the time complexity of the model, especially in comparison to other anomaly detection methods? Does the model’s complexity grow linearly, quadratically, or exponentially with respect to the number of variates and time steps?
3. What specific challenges did you encounter with the datasets used, particularly regarding class imbalances or missing values? How does the model handle such issues in practice, and would it perform well in real-world data with similar characteristics?
4. Can you elaborate on the impact of different time spans and sampling frequencies (e.g., SWaT vs. SMD) on the model’s performance? Does the model perform consistently across datasets with different temporal characteristics?
5. How interpretable is the model? Can you explain the role of the attention mechanism in anomaly detection? Are there any methods in place to visualize or explain the decision-making process of MtsCID?

**Reviewer Confidence:**

4: The reviewer is certain that the evaluation is correct and very familiar with the relevant literature

**Scope:**

3: The work is somewhat relevant to the Web and to the track, and is of narrow interest to a sub-community

---

### Official Review · Reviewer_Qyor · 2024-12-02

**Novelty:** 5
**Technical Quality:** 5

**Review:**

Strong points:
1.	The research issue is interesting.
2.	The datasets for evaluation are adequate for the task.
3.	The paper is worth reading.


Weak points:
1.	Some technological details need further explanations or clarification.
2.	Some symbol details need to be unified.
3.	Some results may require further explanation.

**Questions:**

Detailed comments:
1. In section 2.3, “with F representing the number of frequency components”. The “F” has not been mentioned before. It may be the second dimension “f” in “H”. If it is, please use the same symbol.
2. In Figure 1, some symbols do not show what operations they represent. If possible, please provide a legend.
3. In Section 2.3, the author converts the generated Z∈ RB×L×d into patches of different sizes {Zp1, …, Zpm} in a channel-independent manner. So what is the relationship between ‘m’ and ‘L’? Does it satisfy m=L? Or is it something else?
4. On the GECCO dataset, the MtsCID proposed by the authors achieved a performance that surpassed the second-best baseline by 42.12%. This is significantly higher than the improvement on other datasets. Can this phenomenon be explained?

**Reviewer Confidence:**

3: The reviewer is confident but not certain that the evaluation is correct

**Scope:**

4: The work is relevant to the Web and to the track, and is of broad interest to the community

---

### Official Review · Reviewer_XNcv · 2024-12-02

**Novelty:** 3
**Technical Quality:** 4

**Review:**

Strengths:

S1. The problem of multivariate time series anomaly detection is of important impact in real applications.

S2. The presentation of both the proposed approach and the evaluation is well structured.

Weakness:

W1. Although there is Section 5.1 that discussed two key reasons why MtsCID works, it is still not clear to me what is the key design of the whole architecture that makes it work. The architecture design seems to be driven by two main motivations, (1) the intra- and inter-variable dependencies and (2) the salient features. According to Section 5.1, it is only motivation (2) that plays the main role, which however is not the main part of the architecture as in Figure 1.

W2. Motivation (1) seems to have been largely invested in the general time series analysis literature. p-i module seems to be a new design. However, its fixed interaction patterns require better intuition explanation on why it is sufficient to capture the interactions.

W3. Motivation (3) is not obvious to me what are the “silent” features or “silent” dependencies that cannot be captured in temporal space but have to be in the frequency. It would be very beneficial if the authors could provide a detailed examples.

W4. In addition, there is no strong connections between these two motivations. Why the two-branch design requires “silent” features from the frequency domain, or vice versa?

Revision items:

R1. It is not clear what are the inputs for fc-Transformer, i.e., are they features of individual timestamps, or patches? what are the size of them?

R2. The notations in Figure 2 are not visible. There are similar problems in Figure 3. I suggest the authors to make the fonts the same size as the main text. Figure 1 can be made smaller to save some space.

R3. What are the thresholds for the ASore, and how to chose it for a specific dataset?

**Questions:**

Please discuss the points raised in W1-W4 and R1-R3. In addition,

Q1. fc-Transformer operates in the frequency features. Can these interactions in the frequency domain also be referred to as temporal dependencies? This is not obvious to me.

Q2. Equation in Section 2.6.2 seems to be a pure regularization terms? There seems to be no targets to be learned.

**Reviewer Confidence:**

4: The reviewer is certain that the evaluation is correct and very familiar with the relevant literature

**Scope:**

4: The work is relevant to the Web and to the track, and is of broad interest to the community